# Tat-hspb1 Suppresses Clear Cell Renal Cell Carcinoma (ccRCC) Growth via Lysosomal Membrane Permeabilization

**DOI:** 10.3390/cancers14225710

**Published:** 2022-11-21

**Authors:** Lin Zhang, Guang-Zhi Jin, Dong Li

**Affiliations:** 1Departments of Urology, Tongren Hospital Shanghai Jiao Tong University School of Medicine, Shanghai 200336, China; 2Hongqiao International Institute of Medicine, Tongren Hospital Shanghai Jiao Tong University School of Medicine, Shanghai 200336, China

**Keywords:** renal cancer, peptide, lysosomal membrane permeabilization (LMP), apoptosis

## Abstract

**Simple Summary:**

In this study, we discovered a novel endogenous peptide derived from HSPB1 protein through peptidomic analysis of human renal clear cell carcinoma and adjacent normal tissues. We generated a new peptide by conjugating this HSPB1-derived peptide with the HIV-Tat, named Tat-hspb1. We found that Tat-hspb1 could inhibit the proliferation and migration of ccRCC cells. Furthermore, Tat-hspb1 could induce lysosomal membrane permeabilization (LMP) and apoptosis of ccRCC cells while being less cytotoxic to normal epithelial cells. Tat-hspb1 may be a potential therapeutic agent for renal cancer.

**Abstract:**

Clear cell renal cell carcinoma (ccRCC) is the most prevalent kidney cancer, of which the incidence is increasing worldwide with a high mortality rate. Bioactive peptides are considered a significant class of natural medicines. We applied mass spectrometry-based peptidomic analysis to explore the peptide profile of human renal clear cell carcinoma and adjacent normal tissues. A total of 18,031 peptides were identified, of which 105 unique peptides were differentially expressed (44 were up-regulated and 61 were down-regulated in ccRCC tissues). Through bioinformatic analysis, we finally selected one peptide derived from the HSPB1 protein (amino acids 12–35 of the N-terminal region of HSPB1). Next, we fused this peptide to the HIV-Tat, generated a novel peptide named Tat-hspb1, and found that Tat-hspb1 inhibited ccRCC cells’ viability while being less cytotoxic to normal epithelial cells. Furthermore, Tat-hspb1 induced apoptosis and inhibited the proliferation and migration of ccRCC cells. Furthermore, we demonstrated that Tat-hspb1 was predominantly localized in lysosomes after entering the ccRCC cell and induced lysosomal membrane permeabilization (LMP) and the release of cathepsin D from lysosomes. Taken together, Tat-hspb1 has the potential to serve as a new anticancer drug candidate.

## 1. Introduction

Renal cell carcinoma (RCC) was diagnosed in more than 430,000 people and associated with nearly 180,000 deaths worldwide in 2020. Incidence predominates in men, with the male-to-female ratio being approximately 1.7:1.0 [1]. Approximately 25% of RCC patients present with advanced-stage disease at initial diagnosis, and in patients with localized RCC, nearly 30% will relapse and develop metastasis after tumor resection [2,3]. Common solid renal cell carcinomas include clear cell RCC, papillary RCC, and chromophobe RCC. Clear cell renal cell carcinoma (ccRCC) is the most prevalent histological subtype of kidney cancer and accounts for the majority of metastatic diseases [4]. Targeted therapy used to be the main medical management of metastatic ccRCC due to its high resistance to conventional chemo- and radiotherapies, with targeted therapeutic agents mainly targeting the VEGF signaling axis, mTOR pathway, and tyrosine kinases [5]. According to the latest guidelines on renal cell carcinoma, three immune checkpoint inhibitor (ICI)-based combination therapies of pembrolizumab plus lenvatinib, nivolumab plus cabozantinib, and pembrolizumab plus axitinib were recommended as the new standard of care in all IMDC risk groups. For those who cannot tolerate immune checkpoint inhibitors, targeted therapies are alternative treatment options [6]. Despite having been successfully used for the treatment of metastatic ccRCC, therapy response varies, and severe side effects are prevalent. Combination therapy does improve the prognosis in patients with RCC, but long-term exposure to ICIs can cause resistance owing to mechanisms including neoantigen loss, defects in antigen presentation, alternative immune checkpoints, etc. Furthermore, the incidence of adverse events such as proteinuria and rash increased compared with monotherapy [7]. Furthermore, the limited number of targeted pathways in ccRCC therapy leaves non-responding patients with few therapeutic options. Therefore, agents with novel mechanisms of action deserve attention. 

There are several new treatment options for renal cell carcinoma, which can be divided into five groups: Nonbiologics, small-molecule drugs, biologics, immunomodulatory therapies, and peptide drugs. Peptide drugs have shown promising potential in cancer therapy. For instance, AMG-386 is a peptide drug targeting Ang1 and Ang2 and has shown antitumor activity in treating solid tumors with low toxicity. Currently, there are two active clinical trials investigating this drug as monotherapy and also as a combination with small-molecule inhibitors or anti−PD-1 immunotherapy [8]. Therefore, we focus our attention on peptides.

Peptides are short-chain amino acids connected by amide bonds, with a length of less than 50 amino acids. Many peptides are functional fragments of natural proteins. Thus, compared with traditional targeted therapeutic drugs, peptides have advantages such as remarkable potency, excellent selectivity, and fewer side effects [9,10,11]. Peptides are important bioactive molecules, which play a vital role in transmitting signals and regulating metabolism. For example, peptide p28 derived from azurin inhibits angiogenesis and tumor growth by inhibiting downstream phosphorylation of FAK and Akt [12]. A fusion peptide inhibits proliferation and induces apoptotic death of primary fibroblasts and preleukemic stem cells [13]. Tat–beclin1, an autophagy-inducing peptide, may have the potential for the prevention and treatment of a broad range of human diseases [14]. These studies provide a strong body of evidence supporting the anti-tumor potential of peptides. 

The term ‘peptidomics’ was coined at a scientific meeting organized by Micromass in the late 1990s, which refers to a high-throughput, direct measurement of the structural characteristics of endogenous peptides in a given biological sample [15,16]. Peptidomics is primarily aimed at elucidating the exact form of each peptide detected in the sample, which reflects proteolytic cleavage, other post-translational modifications, and the coexistence of different forms of peptides from the same precursor [17,18]. With the continuous development of liquid chromatography (LC) and mass spectrometry (MS) technologies, the quantitative ability of peptidomics has become more and more reliable. Differential and quantitative peptidomics can quantitatively analyze peptides in specific cell types, tissues, or disease states, revealing the possible biological functions of peptides [19,20,21]. 

In the present study, we performed a comparative peptidomic analysis of human renal clear cell carcinoma and para-cancer tissues to explore the role of endogenous peptides that were involved in oncogenesis by using quantitative liquid chromatography/mass spectrometry (LC/MS). Through systematic screening, we designed and synthesized a cell-permeable peptide named Tat-hspb1, composed of the HIV-1 Tat protein transduction domain (PTD) [22] attached to the amino acid derived from the endogenous protein HSPB1. Here, we initially investigated the anti-tumor effect of Tat-hspb1 on human renal cell carcinoma cells. The cell viability assay indicated that Tat-hspb1 induced a dose-dependent loss of viability in RCC cells with less cytotoxicity to normal epithelial cells. We found that Tat-hspb1 inhibited proliferation and migration while inducing apoptosis of renal cancer cells. Furthermore, our findings show that Tat-hspb1 was mainly localized in lysosomes, inducing lysosomal membrane permeabilization (LMP) and the release of cathepsin D from lysosomes. Thus, our findings suggest that Tat-hspb1 may provide a novel therapy for human renal cell carcinomas.

## 2. Materials and Methods

### 2.1. Sample Collection

Regarding the collection of clear cell renal cell carcinoma (ccRCC) patient tissue samples, three paired ccRCC and adjacent non-tumor tissue samples were collected from Tongren Hospital of Shanghai Jiao Tong University School of Medicine, and all of the patients involved were age matched. The patients were informed about the research and signed medical informed consent documents, and this study was approved by the Ethical Committee of Tongren Hospital of Shanghai Jiao Tong University School of Medicine. No local or systemic treatments were provided to these patients before surgery. All tissues were quickly collected after surgery and stored in liquid nitrogen for preparation and peptide extraction.

### 2.2. Peptide Extraction and Identification

The tissue samples were cut into small pieces, heated, and boiled in water for 10 min (1 g:3 mL). The tissues were oscillated in a grinding machine with glacial acetic acid and acetonitrile at 50 HZ at 4 °C for 10 min. After centrifugation at 4 °C at 12,000× *g* for 30 min, the supernatant was transferred to EP tubes and lyophilized. Then an 80% acetone solution was added and oscillated for 2 min at 4 °C. The sample was centrifuged and lyophilized again as described above, samples were redissolved in 0.1% TFA and desalted and concentrated by C18 solid-phase extraction, and, finally, lyophilized in vacuo. Samples were labeled with the TMT reagent, and LC-MS/MS analysis was performed simultaneously to compare the abundance of peptides in different samples. The samples were analyzed three times for each sample. Maxquant software was used to analyze the peptides’ MS/MS spectra. Peptides with a fold change larger than 2 with a *p*-value < 0.05 were selected as differentially expressed proteins. 

### 2.3. Bioinformatics Analysis and Peptide Synthesis

We used an online computational tool (https://web.expasy.org/protparam/, accessed on 12 August 2021) to analyze the physical properties of the peptides, such as the isoelectric point (pI), molecular weight (Mw), grand average of hydropathicity, estimated half-life, etc. Gene ontology (GO) analysis and Reactome pathways analysis were carried out to clarify the potential function of the peptide-related precursor proteins. The interaction network function of the identified peptides’ precursor proteins was analyzed using STRING (https://string-db.org/, accessed on 25 August 2021). Peptide Tat-hspb1 (RKKRRQRRR-RGPSWDPFRDWYPHSRLFDQAFGL) and Tat-hspb1-Flag (RKKRRQRRR-RGPSWDPFRDWYPHSRLFDQAFGL-Ahx-DYKDDDK) were synthesized by Science Peptide Biological Technology (Shanghai, China) through the solid-phase method. The purity of the peptide was beyond 95% detected by the HPLC-MS method. It was preserved via freeze-drying at −20 °C before being immediately dissolved in double-distilled water for cell treatment in vitro.

### 2.4. Cell Culture and Reagents

Human RCC cell lines 786-O, A498, Caki-1, and human umbilical vein endothelial HUVEC cells were obtained from the cell bank of the Chinese Academy of science. Human renal tubular epithelial cell HKC cells were purchased from Procell (Wuhan, China). All cells were cultured in the Dulbecco modified essential medium (Gibco, Grand Island, NY, USA) or the Roswell Park Memorial Institute (RPMI) 1640 medium (Gibco, Grand Island, NY, USA) supplemented with 10% fetal bovine serum (Gibco, Grand Island, NY, USA) and 1% P/S (Gibco, Grand Island, NY, USA) at 37 °C in 5% CO_2_. All cell lines were tested for mycoplasma contamination and were validated by short tandem repeat (STR) polymorphism analysis.

Antibody and reagents: anti-Caspase-3 (ProteinTech Group, Chicago, IL, USA, 19677-1-AP, 1:1000), anti-Caspase-8 (ProteinTech Group, Chicago, IL, USA, 13423-1-AP, 1:500), anti-Caspase-9 (ProteinTech Group, Chicago, IL, USA, 10380-1-AP, 1:500), anti-β-Actin (Abcam, Cambridge, UK, ab8226, 1:5000), anti-α-Tubulin (Abcam, Cambridge, UK, ab7291, 1:5000), CoraLite488-conjugated Goat Anti-Mouse IgG(H+L) (ProteinTech Group, Chicago, IL, USA, SA00013-1, 1:1000), CoraLite594-conjugated Goat Anti-Rabbit IgG(H+L) (ProteinTech Group, Chicago, IL, USA, SA00013-4, 1:1000), anti-LAMP1 (CST, Boston, MA, USA, 9091T, 1:250), anti-Flag (Immunoway, Plano, TX, USA, YM3001, 1:250), anti-Cathepsin D (Abcam, Cambridge, UK, ab75852, 1:150), Goat anti-Rabbit IgG (H + L), HRP conjugate (Abcam, Cambridge, UK, ab97051, 1:10,000), Goat anti-Mouse IgG (H + L), HRP conjugate (Abcam, Cambridge, UK, ab6789, 1:10,000). Z-VAD-FMK, Necrostatin-1, CQ, Ac-FLTD-CMK, and Pepstatin A were purchased from MedChemExpress (NJ, USA).

### 2.5. Cell Viability Measurement

The CCK8 assay was used to evaluate the effect of Tat-hspb1 on cell viability. 786-O, A498, Caki-1, HUVEC, and HKC cells were inoculated in 96-well plates (5–8 × 10^3^ cells/well). Twenty-four hours later, cells were treated with Tat-hspb1 (0–100 μg/mL). After treatment with peptides for 0 h, 22 h, and 46 h, the CCK8 reagent was added to each well and incubated for 2 h at 37 °C. The absorbance of each well at 450 nm was measured by the Multiskan FC Microplate Reader (Thermo Scientific, Waltham, MA, USA). Each experiment was performed in triplicate.

### 2.6. Wound Healing Assay

Caki-1 and 786-O cells were seeded in 12-well plates (5.0 × 10^5^ cells/well). Forty-eight hours after inoculation, the cell layer was scratched using a sterile 200 μL pipette tip when adherent cells filled up the well, and floating cells were removed by washing with 1 × PBS. Next, RPMI-1640 containing 1% FBS and different concentrations of Tat-hspb1 (0, 10, 20, and 30 μg/mL) was added to each well. The wound was captured using a Nikon Inverted Research Microscope Eclipse Ti microscope at 40× magnification at 0 h, 9 h, and 24 h after scratching, and the wound area was assessed by quantitative analysis using ImageJ software.

### 2.7. Colony Formation Assay

786-O and Caki-1 cells were seeded in 6-well plates (2000 cells/well) with three replicate wells in each group. Cells were treated with different concentrations of Tat-hspb1 (0, 10, 20, and 30 μg/mL) and incubated at 37 °C for one week. The cells were fixed with methanol at room temperature for 20 min and then stained with crystal violet (0.2% *w*/*v* in methanol) for 15 min and photographed. Colonies were counted under a microscope. 

### 2.8. Microscopy Imaging of Cell Death

To observe cell death morphology, Caki-1 and 786-O cells were seeded in 12-well plates (5.0 × 10^5^ cells/well) for static image capture. Static bright field images of cells were captured using a Nikon Inverted Research Microscope Eclipse Ti microscope at 100× magnification at 0 h, 2 h, 4 h, and 8 h after Tat-hspb1 (80 μg/mL) treatment. The image pictures were processed using ImageJ, and all images shown are representative of at least three randomly selected fields.

### 2.9. Flow Cytometry Analysis

To access apoptosis on Caki-1 and 786-O cells treated with Tat-hspb1, flow cytometry was carried out by an Annexin V/PI double-staining assay. First, Caki-1 and 786-O cells were seeded in 6-well plates (8 × 10^5^ cells/well) and incubated overnight. Then, cells were washed with 1 × PBS, and a culture medium containing gradient concentrations of Tat-hspb1 (0, 40, and 80 μg/mL) was added to each well. After 24 h incubation at 37 °C, the cells were harvested, centrifuged, washed, resuspended in 1X binding buffer, stained with FITC Annexin V and PI (BD Biosciences, San Jose, CA, USA), and incubated for 15 min at room temperature in the dark. Finally, cell apoptosis was detected on a FACS auto flow cytometer (BD Biosciences) and the data were analyzed by FlowJo software.

### 2.10. Western Blot Analysis

After treatment, cells were lysed with RIPA buffer (Beyotime, Shanghai, China) plus 1 mM PMSF and the Protease inhibitor cocktail (Beyotime, Shanghai, China). The protein concentration was determined using the BCA Protein Assay (Thermo Scientific, Waltham, MA, USA). Total proteins (20 μg) were separated by SDS-PAGE (10%, 12.5%) and subsequently transferred to polyvinylidene difluoride (PVDF) membranes (Millipore Corporation, Billerica, MA, USA). The membranes were blocked in 5% skimmed milk and incubated at room temperature for 1 h, followed by incubation with specific primary antibodies overnight at 4 °C. The blots were washed with TBST three times and then probed with HRP-conjugated secondary antibodies for 2 h at room temperature. The membranes were washed as described previously, and the blots were visualized on the Tanon-5200 Chemiluminescent Imaging System (Tanon Science & Technology, Shanghai, China). β-Actin and α-Tubulin were used to ensure equivalent loading of the whole cell protein.

### 2.11. AO Staining and Immunofluorescence

786-O cells were seeded in 12-well plates (5.0 × 10^5^ cells/well) and incubated overnight. Then cells were washed with 1 × PBS, and a culture medium containing Tat-hspb1 (80 μg/mL) was added to each well. After 2 h incubation, cells were washed with 1 × PBS and stained with Acridine orange (AO; Sigma-Aldrich, St. Louis, MI, USA) at 50 μg/mL in a complete medium for 15 min at 37 °C. The changes in red and green fluorescence were visualized on the Nikon Inverted Research Microscope Eclipse Ti microscope at 200× magnification. Pictures were analyzed by ImageJ software.

786-O cells were inoculated on glass coverslips at the bottom of 6-well plates at a density of 5.0 × 10^5^ cells/mL and treated with Tat-hspb1 (80 μg/mL). After treatment, coverslips were fixed in ice-cold methanol for 20 min and then permeabilized with a 0.3% Triton X- 100 solution for 10 min at room temperature. After washing with PBS for 3 × 5 min, cells were then blocked for 30 min in 5% bovine serum albumin (BSA) diluted with 1 × PBS. The cells were incubated with the anti-cathepsin D primary antibody (Abcam, USA) at a 1:150 dilution overnight at 4 °C. After washing with 1 × PBS, the cells were further incubated with the CoraLite488-conjugated secondary antibody (ProteinTech Group, Chicago, IL, USA) for 1 h, then stained with DAPI for 5 min at room temperature. Stained cells were visualized and photographed using the Leica SP8 confocal scanning microscope at 200× magnification. 

786-O cells were treated with Tat-hspb1-Flag (80 μg/mL) for 0 h, 1 h, 2 h, and 4 h, and then fixed and blocked as described above. Cells were incubated with anti-LAMP1 (CST, Boston, MA, USA) and anti-Flag (Immunoway, Plano, TX, USA) primary antibodies at a 1:250 dilution overnight at 4 °C. After washing with 1 × PBS, the cells were further incubated with the CoraLite488 or CoraLite594-conjugated secondary antibody (ProteinTech Group, Chicago, IL, USA) for 1 h and then stained with DAPI for 5 min at room temperature. Stained cells were visualized and photographed using the Leica SP8 confocal scanning microscope at 200× magnification. 

### 2.12. Statistical Analysis

All experiments, unless differently indicated, were performed at least three times. All data were expressed as the arithmetic mean and standard deviation (S.D.) of measurements. Statistical analysis was conducted using GraphPad Prism 9.0 software. Student’s t-test or the one-way analysis of variance (ANOVA) were used for statistical significance of the differences between treatment groups. A value of *p* < 0.05 was considered statistically significant.

## 3. Results

### 3.1. Peptidome Characterization of ccRCC and Adjacent Normal Tissue and Bioinformatic Analysis of Differentially Expressed Peptides

We collected paired ccRCC and adjacent non-tumor tissue samples from three patients (Appendix A). A total of 18,031 peptides from 2419 precursor proteins were detected in both ccRCC and adjacent normal tissues. As shown in the volcano plot (Figure 1A), we identified 115 differentially expressed peptides from 83 types of precursor proteins (fold change >2 or <0.5, *p* < 0.05). By comparing the sequences of these differentially expressed peptides to proteins, most peptides only correspond to one precursor protein, and a portion of peptides correspond to several precursor proteins. After excluding the ‘non-unique’ peptides, we finally obtained 105 differentially expressed peptides, among which 44 peptides from 31 precursor proteins were up-regulated in ccRCC tissues, and 61 peptides from 49 precursor proteins were down-regulated in ccRCC tissues (Figure 1B and Appendix A). Many peptides’ biological function is similar or opposite to their precursor proteins [23]. To further investigate the latent function of the differentially expressed peptides, gene ontology (GO) functional annotation and Reactome pathway enrichment analysis of the precursor proteins of differentially expressed peptides were performed. Enzyme binding and RNA binding were the most highly enriched molecular functions (Figure 1C). Localization and establishment of localization were the most highly enriched biological processes (Figure 1D). The intracellular part and organelle part were the most highly enriched cellular components (Figure 1E). Reactome pathway analysis showed that the precursor proteins of differentially expressed peptides were enriched in the pathways of metabolism, immune system, innate immune system, cellular response to stress, etc. (Figure 1F). We subsequently analyzed the protein–protein interaction (PPI) networks of these differentially expressed peptides’ precursor proteins according to STRING (https://string-db.org/, accessed on 25 August 2021). The results indicated that the top 10 interaction networks existed in the proteins GAPDH, EEF2, SOD1, PKM, HNRNPA2B1, ATP5I, CCT6A, HSPB1, RPS28, and UBA52 at the core position (Figure 1G).

### 3.2. Tat-hspb1 Causes Loss of Viability in Renal Cancer Cells and Less Cytotoxic Effect to Normal Epithelial Cells

The node degree is the number of interactions that a protein has in the PPI network. Hub nodes are those proteins with the highest degree, which have key responsibilities in maintaining network stability and processing signal propagation, reflecting possible significance in the oncogenesis of renal cancer [24]. Logically, down-regulated peptides in ccRCC tissues may have antitumor properties. Thus, to find potential bioactive peptides, we outlined List 1, namely, the top 10 hub precursor proteins in PPI networks; List 2, namely, precursor proteins belonging to the category ‘enzyme binding’ (BP); and List 3, namely, precursor proteins down-regulated in ccRCC tissues. Their intersection with protein-HSPB1 was visualized via the Venn diagram tool (Figure 2A). There are two peptides corresponding to protein HSPB1(Table 1), and we selected the one consisting of 24 amino acids with a higher absolute fold change ratio and smaller *p*-value. A membrane permeabilizing peptide is one that creates a pathway that enables the passage of polar molecules across the lipid bilayer membranes. A small peptide with the sequence RKKRRQRRR derived from the transduction domain of the HIV Tat protein has been successfully shown to deliver a variety of cargo, from small particles to proteins, peptides, and nucleic acids [25]. Thus, we generated an HSPB1-derived peptide conjugated to Tat (Tat-hspb1) and demonstrated its bioactive function in different cell lines. To test the effect of Tat-hspb1 on the viability of renal cancer cells, 786-O, Caki-1, and A498 cell lines were exposed to different concentrations of Tat-hspb1 for 24 h and 48 h. The CCK-8 assay showed that the viability of renal cancer cell lines was significantly reduced by Tat-hspb1 in a dose-dependent manner compared with that of the saline control (Figure 2B–D). The IC50 values of Tat-hspb1 for 24 h were 60.5 μg/mL for A498, 55.9 μg/mL for 786-O, and 62.4 μg/mL for Caki-1. Furthermore, we compared the effect of Tat-hspb1 with Sunitinib (Appendix A), and the results suggested that they have similar antitumor ability. More importantly, Tat-hspb1 showed little effect on human renal tubular epithelial cells HKC with an IC50 value of 84.9 μg/mL and human umbilical vein endothelial cells HUVEC with an IC50 value of 93.7 μg/mL (Figure 2E,F). These multiple lines of evidence suggest that Tat-hspb1 has a pro-death effect on renal cancer cells and is less cytotoxic to normal cells.

### 3.3. Tat-Hspb1 Inhibits the Proliferation and Migration of Renal Cancer Cells In Vitro

To explore the biological function of Tat-hspb1 in ccRCC cell lines, we performed a colony formation assay and a wound-healing assay. We observed that the number of colonies in 786-O and Caki-1 cells was significantly reduced after treatment with low gradient Tat-hspb1 concentrations compared with the control group (Figure 3A). Moreover, after exposure to different Tat-hspb1 concentrations, ranging from 10 to 30 μg/mL, 786-O and Caki-1 cells’ migration velocity was significantly decreased compared with the control group (Figure 3B). These results suggest that Tat-hspb1 inhibits the proliferation and migration of renal cancer cells.

### 3.4. High Concentration of Tat-hspb1 Induces Apoptosis in Renal Cancer Cells

During the CCK-8 assay, we found a great deal of floating debris or dead cells in high-concentration Tat-hspb1 groups. To identify the cell death induced by Tat-hspb1, 786-O and Caki-1 cells were incubated in 12-well plates and observed via a microscope in the bright field at different time points after treatment with high Tat-hspb1 concentrations (80 μg/mL). Compared with the control group, there were increasing small bubbles and black granules in cells at the early stage of induction, adherent cells were reduced, and some floating dead cells in shiny spheroids shapes appeared in the late stage of Tat-hspb1 treatment (Figure 4A). To understand the pattern of Tat-hspb1-induced cell death, Annexin V/PI positive cells were detected by flow cytometry after 24 h exposure to Tat-hspb1 (Figure 4B). There was a significant increase in Annexin V- and PI-positive cells with increasing concentrations of Tat-hspb1. Apoptosis is a conserved process, which can be induced both intrinsically and extrinsically and culminates in the activation of caspases. Both intrinsic and extrinsic pathways finally led to the proteolytic maturation of executioner caspases, mainly CASP3 [26,27]. As shown in Figure 4C and Appendix A, the expression levels of cleaved-caspase-3, cleaved-caspase-8, and cleaved-caspase-9 were increased in response to Tat-hspb1 treatment. Altogether, these results suggest that Tat-hspb1 can induce caspases-dependent apoptosis.

### 3.5. Tat-hspb1 Was Predominantly Localized in Lysosomes, Inducing Lysosomal Membrane Permeabilization and the Release of Cathepsin D from Lysosomes

To further investigate the mechanism underlying Tat-hspb1-induced cell death, we pretreated 786-O and Caki-1 cells with certain types of inhibitors for 1 h before Tat-hspb1 exposure. As shown in Figure 5A, pepstatin A (cathepsin D inhibitor) [28] and necrostatin-1 (RIPK1 and cathepsin D inhibitor) [29,30] significantly rescued Tat-hspb1-induced cell death, while other inhibitors, including Z-VAD-FMK (pan-caspase inhibitor) [31], CQ (autophagy inhibitor) [32], and Ac-FLTD-CMK (pyroptosis inhibitor) [33] had little or no effect on Tat-hspb1-induced cell death. This suggests that Tat-hspb1 may induce lysosomal membrane permeabilization (LMP) and the subsequent release of lysosomal enzymes, especially cathepsins, into the cytosol. To investigate whether Tat-hspb1 was concentrated in the lysosome, we constructed Flag-conjugated Tat-hspb1. We cultured 786-O cells in vitro and performed immunofluorescence to examine the subcellular colocalization of Tat-hspb1 with lysosome-associated protein 1 (LAMP1) at different time points. As shown in Figure 5B, more Tat-hspb1 entered cells as time went on, and the subcellular localization gradually lacked its initial discrete punctate appearance. Furthermore, an increasing proportion of Tat-hspb1 colocalized with endogenous LAMP1, suggesting that Tat-hspb1 was prominently localized in lysosomes. To investigate whether LMP actually occurred in ccRCC cells after Tat-hspb1 treatment, we first stained untreated and Tat-hspb1-treated 786-O cells with acridine orange (AO) [34], which accumulated in the lysosomes and exhibited red fluorescence in the acid compartment of the intact lysosomes but emitted green fluorescence when the lysosomal membrane integrity was disrupted. In the NC group, cells showed bright lysosomal red and low cytosolic green fluorescence, indicating intact lysosomal membranes. However, red fluorescence was remarkably reduced in Tat-hspb1-treated cells (Figure 5C), indicative of LMP. We then performed immunofluorescence analysis to determine the subcellular relocation of cathepsin D. Immunofluorescence staining was detected in the punctate pattern representing intact lysosomes where cathepsin D normally localized in. After Tat-hspb1 treatment, green fluorescence became diffuse (Figure 5D), suggesting that cathepsin D was relocated in the cytosol, in accordance with the concept of LMP. Collectively, these results indicate that Tat-hspb1 is involved in lysosome-dependent cell death.

## 4. Discussion

Renal cell carcinoma accounts for approximately 2–3% of all malignant tumors worldwide, of which the incidence increases annually. The pathogenesis of kidney cancer is not well understood, likely related to obesity, smoking, and hypertension. Localized RCC can be successfully treated with partial or radical nephrectomy or active surveillance, whereas metastatic RCC is refractory to conventional chemotherapy [35,36]. At present, targeted agents are approved as first-line options for the treatment of metastatic RCC. However, those drugs are associated with different toxicities. Pazopanib may induce hand-foot syndrome and liver function test abnormalities [37] and Bevacizumab can cause Proteinuria and hypertension [38]. Therefore, the search for drugs with high potency and low toxicity remains a challenge.

For cancer therapy, peptides have been widely investigated due to their attractive benefits, including their broad chemical diversity, wide range of targets, good biocompatibility, and safety. Therapeutic peptides can target specific cell types, signaling pathways, or cancer-causing proteins [39,40]. Some peptides display inhibitory activities directly in tumor cells. For instance, Peptide NuBCP-9 exerts great anticancer activity and is harmless to normal cells [41]. Furthermore, some peptides can activate immune cells to kill tumor cells, for example, anti-PD1-peptide could block the PD1/PD-L1 pathway and revoke T cell functions, making it possible for the immune system to attack tumor cells [42]. Endogenous peptides are considered multifunctional and have been found to be involved in numerous biological processes. Peptidomics is an emerging branch of proteomics that targets endogenously produced protein fragments [43,44]. Differential and quantitative peptidome analysis of different samples exhibits a differential peptidomic profile. In our study, we performed a peptidomic study of human ccRCC tissues and paired adjacent non-tumor tissues. We identified 105 unique natural peptides from 80 precursor proteins, which were differentially expressed, and shed light on the possible role of endogenous peptides in ccRCC prevention and treatment.

Previous studies have shown that peptides’ function is usually similar or opposite to their precursor proteins [23,45]. Thus, we performed a bioinformatic analysis of differentially expressed peptide precursors. Our GO analysis showed that the molecular function of differential peptide precursors was mainly related to protein binding. Peptides can target proteins and protein–protein interactions with high specificity, inducing selective cell death and preventing a broad range of human diseases [14]. Tumor-promoting peptides are usually expressed at high levels in cancers and enhance the malignant behavior of tumor cells. In order to find promising anti-cancer peptides, we selected peptides that were down-regulated in ccRCC tissues compared with adjacent non-tumor tissues. PPI networks are already proving valuable in providing insight into complex diseases and revealing the functional relationships of different proteins, as proteins within the same disease are more likely to interact or belong to the same functional modules in biological networks [46]. Thus, we chose the top 10 precursor proteins after PPI analysis. Based on those results, we determined the overlap between the three categories and focused on one peptide derived from the HSPB1 protein (amino acids 12–35 of the N-terminal region of HSPB1). HSPB1 belongs to the family of human small heat shock proteins and is responsible for the binding of improperly folded protein substrates and their further transfer to ATP-dependent chaperones or protein degradation machines such as proteasomes or autophagosomes [47,48]. However, due to poor membrane permeability, numerous peptides’ intracellular delivery was impeded. Hence, the hspb1 peptide was conjugated to the protein transduction domain derived from the HIV-1 Tat protein to increase its intracellular delivery efficiency [49]. 

By synthesizing the peptide Tat-hspb1, we verified its function in several cell lines. Our results showed that Tat-hspb1 decreases the cellular viability of RCC cells in a dose-dependent manner, and the IC50 value of Tat-hspb1 for normal human epithelial cells was higher than that of RCC cells, suggesting the possibility of Tat-hspb1 for the treatment of renal cancer. In normal cells, a negatively charged phospholipid is usually present on the inner leaflet of the cell membrane while the outer leaflet is zwitterionic. However, some anionic molecules are present specifically on the membranes of cancer cells, and the asymmetry of the membrane is lost [50]. The PI of Tat-hspb1 is 11.91, so it may explain why cancer cells are more susceptible to Tat-hspb1. Migration and proliferation are characteristics of malignant tumors, resulting in the major mortality of cancer patients [51]. The results of our study showed that Tat-hspb1 inhibited the migration and proliferation of ccRCC cells in a dose-dependent manner, whereas the specific mechanism needs further study. These results indicate that Tat-hspb1 is a potential therapeutic agent for renal cancer.

Previous studies have demonstrated that HSPB1 is an endogenous pleiotropic inhibitor of apoptotic cell death. HSPB1 interferes negatively with apoptosis by binding to cytochrome c released from the mitochondria to the cytosol and prevents the cytochrome-c-mediated interaction of Apaf-1 with procaspase-9. Amino acids 51–88 of the amino-terminal region of HSPB1 play an indispensable role in this process [52,53]. However, our results showed that high concentrations of Tat-hspb1, which is derived from endogenous HSPB1, induce ccRCC cell death in vitro. The results of flow cytometry showed that the apoptosis rate increased as the concentration of Tat-hspb1 increased. We observed distinct apoptotic morphology changes in ccRCC cells after treatment with Tat-hspb1. Apoptosis is characterized by cell shrinkage, DNA fragmentation, and activation of caspases, and it can be subdivided into the intrinsic pathway and the extrinsic pathway. Intrinsic apoptosis is initiated by a variety of perturbations such as DNA damage and reactive oxygen species, and then BCL2 induces mitochondrial outer membrane permeabilization (MOMP) and subsequent cytosolic release of cytochrome c. Cytochrome c binds to APAF1 and caspase-9 to form apoptosome. Activated caspase-9 can catalyze the proteolytic activation of caspase-3. Extrinsic apoptosis is initiated by extracellular stimulus, and then death receptors recruit and activate caspase-8. Caspase-8-dependent proteolytic maturation of executioner caspase-3 causes cell death [54,55,56,57]. Our results showed that the expression of cleaved-caspase-3, cleaved-caspase-8, and cleaved-caspase-9 was up-regulated when ccRCC cells were treated with Tat-hspb1. We conclude that Tat-hspb1 induces ccRCC cell apoptosis in a caspase-dependent manner, but whether Tat-hspb1 directly promotes apoptosis is still unknown.

Lysosome-dependent cell death is a type of RCD demarcated by primary LMP and precipitated by cathepsins, with the optional involvement of MOMP and caspases [27]. The degree of LMP determines the morphological characteristics of cell death. Extensive LMP leads to uncontrolled necrosis and rapid permeabilization of the plasma membrane, while limited LMP can activate the intrinsic apoptosis in a caspase-independent or -independent way [58,59]. Cathepsins are executors of LMP-induced cell death, which can trigger MOMP and apoptosis, and the inhibition of cathepsins, especially cathepsin D and cathepsin B, can confer significant protection against cell death following limited LMP [60]. Cytosolic cathepsins can activate pro-apoptotic Bid protein or apoptotic caspases by cleaving them [61]. Our results showed that Necrostatin-1 and pepstatin A can significantly rescue Tat-hspb1-induced cell death. Pepstatin A is an inhibitor pharmacologically targeting cathepsin D, and necrostatin-1 is a RIPK1 inhibitor, which also has an inhibitory action on cathepsin D [30], suggesting that Tat-hspb1 induces the release of cathepsin D from the lysosome to the cytosol. Based on our immunofluorescence experiment, our study confirmed that the exposure of ccRCC cells to Tat-hspb1 leads to LMP, with the release of cathepsins such as cathepsin D into the cytosol. Taken together, these results imply that the mechanisms of Tat-hspb1-induced cell death are associated with increased LMP and cathepsin release. Still, there are several limitations to our study. First, all experiments were performed in vitro. We need to further assess the in vivo potency of Tat-hspb1. Second, our research only explored one aspect of Tat-hspb1-induced ccRCC cell death, and further studies are needed to systematically investigate the molecules and pathways involved in Tat-hspb1-induced cell death. More importantly, for better clinical translation, chemical modification of Tat-hspb1 may be needed.

## 5. Conclusions

In conclusion, through peptidomic analysis, we first identified an endogenous peptide derived from HSPB1 and generated a novel peptide Tat-hspb1. We showed that Tat-hspb1 can cause ccRCC cell death while being less toxic to normal epithelial cells. Furthermore, Tat-hspb1 inhibits the proliferation and migration of ccRCC cells. Moreover, we demonstrated that LMP and cathepsin D are involved in Tat-hspb1-induced cell death, establishing a basis for the further investigation of Tat-hspb1 cytotoxicity in an in vivo system and providing a potential application in the treatment of renal cancer.

## Figures and Tables

**Figure 1 cancers-14-05710-f001:**
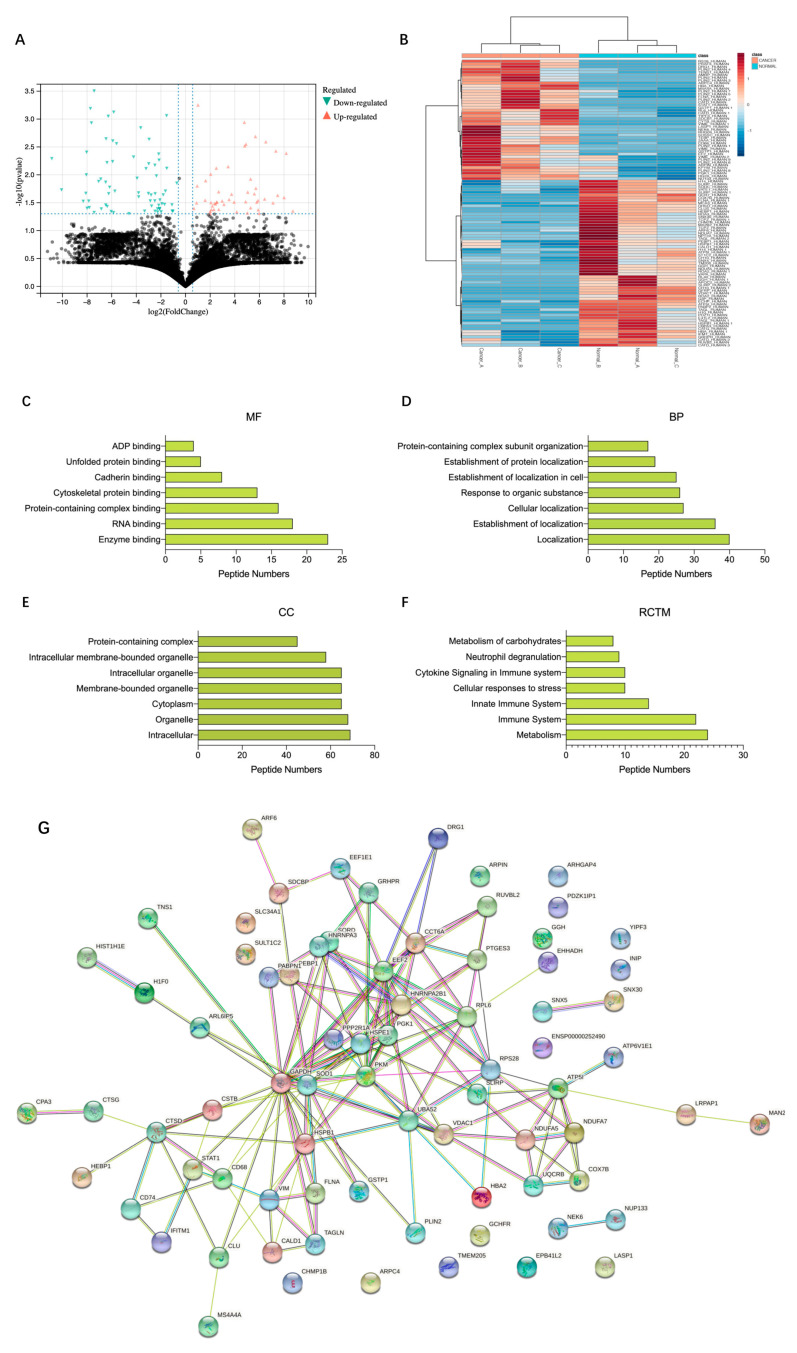
Characteristics of differentially expressed peptides and bioinformatic analysis of precursor proteins from which differentially expressed peptides were derived. (**A**) A volcano plot identified 115 differentially expressed peptides (fold change >2 or <0.5, *p* < 0.05). (**B**) A heat map identified 105 unique differentially expressed peptides. (**C**) Molecular function. (**D**) Biological processes. (**E**) Cellular component. (**F**) Reactome pathway enrichment analysis. (**G**) Protein–protein interaction network analysis based on STRING.

**Figure 2 cancers-14-05710-f002:**
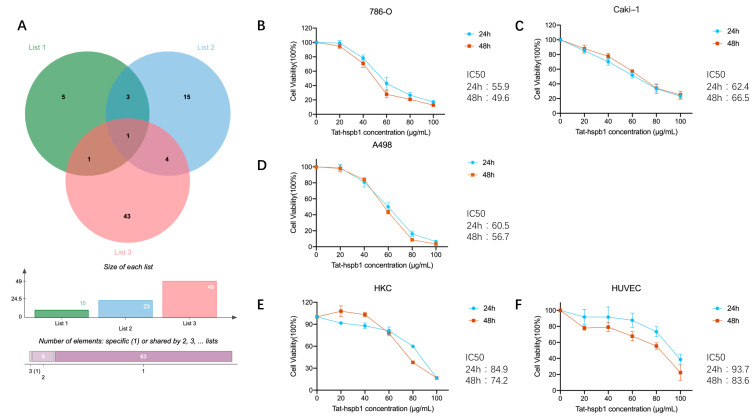
Selection of peptides with potential bioactivity. Cell viability was determined by CCK-8 assays. (**A**) A Venn diagram identified two peptides derived from the same precursor protein HSPB1. (**B**–**D**) Renal cancer cell lines 786-O, Caki-1, and A498 and (**E**,**F**) normal epithelial cells HUVEC and HKC were treated with various concentrations of Tat-hspb1 for 24 h and 48 h or double-distilled water as a control. The results are expressed as the means ± SD of three independent experiments.

**Figure 3 cancers-14-05710-f003:**
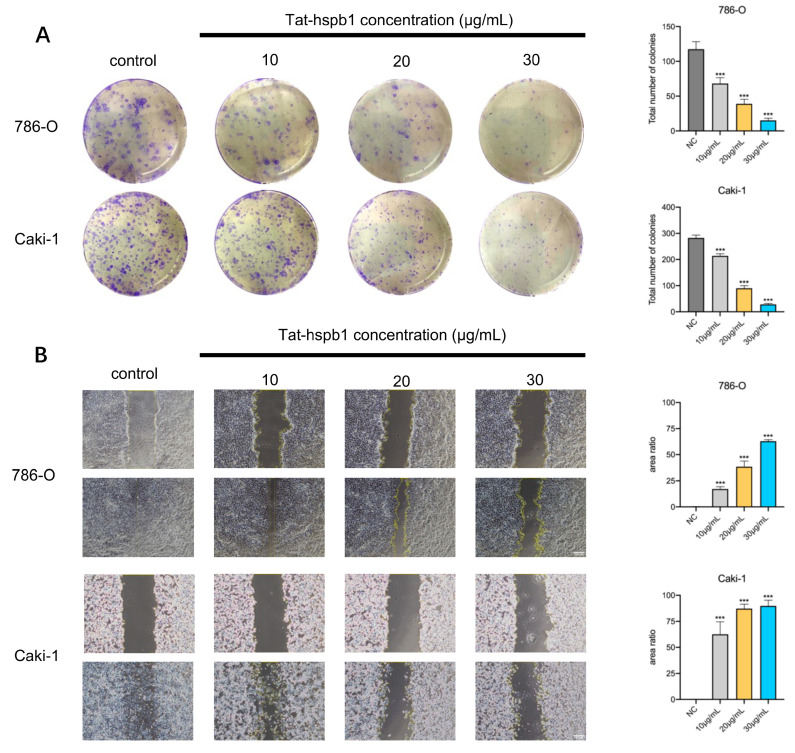
Tat-hspb1 inhibits proliferation and migration of renal cancer cells. (**A**) Representative images and quantification results of colony formation assay of 786-O and Caki-1 cells after treatment with low-gradient concentrations of Tat-hspb1. (**B**) Representative images and quantification results of wound-healing assay of 786-O and Caki-1 cells after treatment with low-gradient Tat-hspb1 concentrations. The results are expressed as the means ± SD of three independent experiments. *** *p* < 0.001.

**Figure 4 cancers-14-05710-f004:**
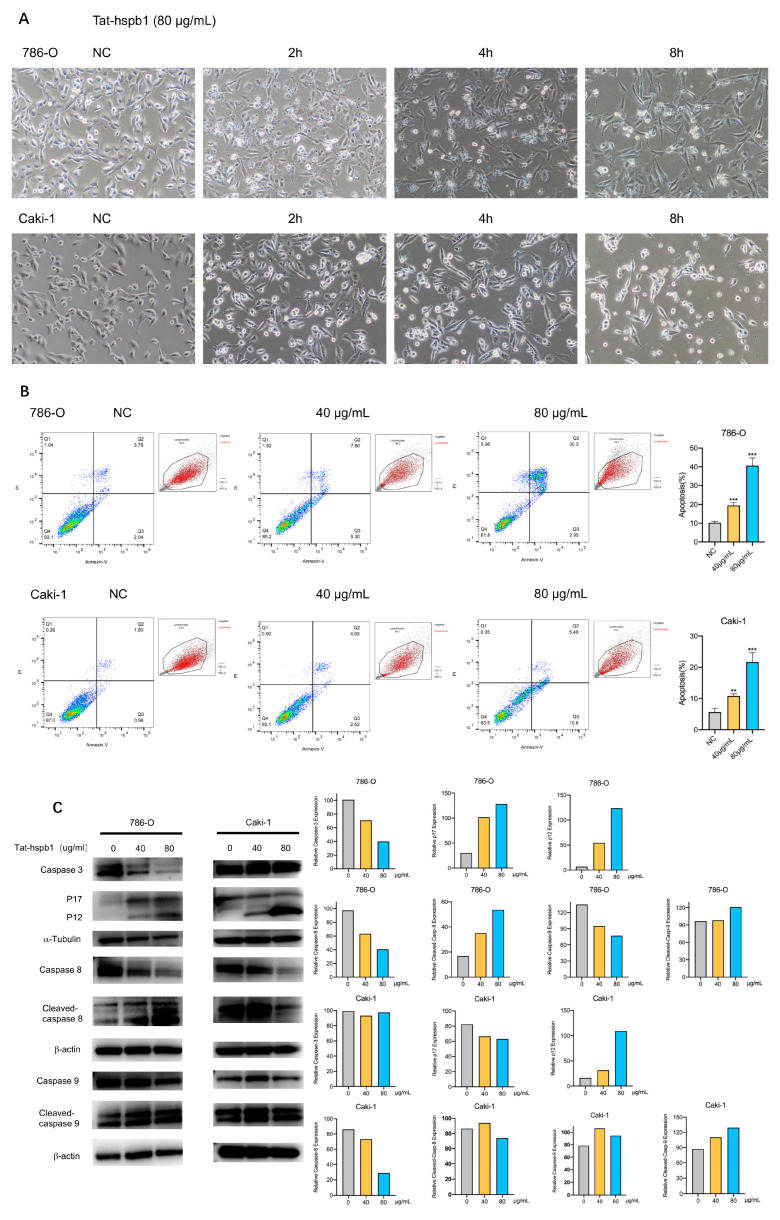
Tat-hspb1 induces apoptosis in renal cancer cells. (**A**) Bright-field photomicrographs show obvious morphological change of 786-O and Caki-1 cells after treatment with Tat-hspb1 (80 μg/mL) or double-distilled water as a control. (**B**) 786-O and Caki-1 cells were treated with the different concentrations of Tat-hspb1 for 24 h, Representative graphs were obtained from cytometry analysis. (**C**) 786-O and Caki-1 cells were treated with the different concentrations of Tat-hspb1 for 24 h; the protein expression of caspase-3, cleaved-caspase-3, caspase-8, cleaved-caspase-8, caspase-9, and cleaved-caspase-9 was determined by Western Blot. The results are expressed as the means ± SD of three independent experiments. ** *p* < 0.01, *** *p* < 0.001.

**Figure 5 cancers-14-05710-f005:**
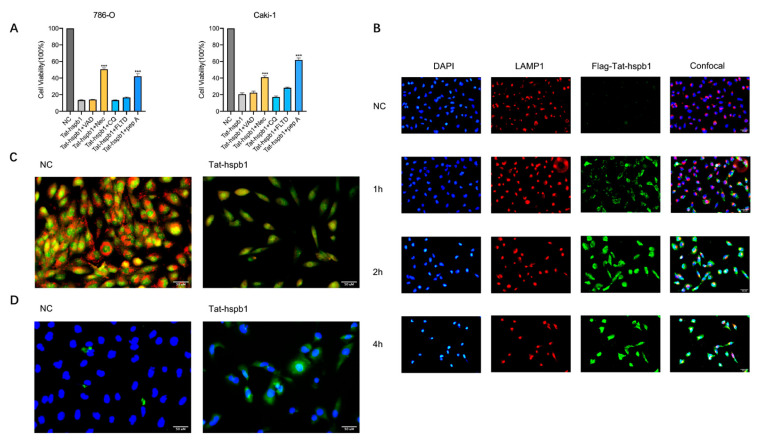
Tat-hspb1 induces lysosomal membrane permeabilization (LMP) in renal cancer cells. (**A**) 786-O and Caki-1 cells were treated with Tat-hspb1 (80 μg/mL) either alone or combined with specific inhibitors, Z-VAD-FMK (80 μM), necrostatin-1 (80 μM), CQ (80 μM), Ac-FLTD-CMK (80 μM), and pepstatin-1 (80 μM) for 24 h, and viability was assessed by CCK-8 assay. (**B**) 786-O cells were treated with Flag-Tat-hspb1 (80 μg/mL) for 1, 2, and 4 h, and double-distilled water was used as a control. Typical confocal images were obtained, where green fluorescence represents Tat-hspb1 and red fluorescence represents LAMP1. (**C**) 786-O cells untreated or treated with Tat-hspb1 (80 μg/mL for 2 h) were stained with acridine orange (AO). (**D**) 786-O cells untreated or treated with Tat-hspb1 (80 μg/mL for 2 h) were stained with the anti-Cathepsin D antibody. The results are expressed as the means ± SD of three independent experiments. *** *p* < 0.001.

**Table 1 cancers-14-05710-t001:** Peptides corresponding to the protein HSPB1.

HSPB1	Log2 FC	*p* Value
PAVAAPAYSRALSRQL	−1.103	0.034
RGPSWDPFRDWYPHSRLFDQAFGL	−3.818	0.021

## Data Availability

All new data have been presented in this paper. There are no further data, but the author welcomes questions and discussion.

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
