# Peer review of "Tat-hspb1 Suppresses Clear Cell Renal Cell Carcinoma (ccRCC) Growth via Lysosomal Membrane Permeabilization"

_cancers, 2022, doi:10.3390/cancers14225710_

Round 1
Reviewer 1 Report
Reviewer comments
Zhang et al described a novel class of peptides that can be used in lieu of chemotherapy and assessed their biological activity in patient derived Renal cell carcinoma cell lines. They found that this peptide serves as good potential target for RCC.
Overall the study fits the scope of Cancers Journal. However, there are a few things that need to be addressed:
1. Introduction is very clear but sometimes the continuity between the statements is missing. For examples: At the end of first paragraph, they do talk about novel therapies for cancer and then they can just include a statement connecting it to the next paragraph where they are talking about Peptides.
It does make sense at the end of that peptide paragraph, why they were talking about it but it would be good to rearrange certain sentences to make it more cohesive.
2. Figure1B: The panels are very small and blurry (not readable), to show the different genes expressed. The figure can be made a bit bigger.
3. Figure 2B-F: The IC50 values that were shown are in mg/ml range which is nothing but mM concentration which is a little higher. Have you done any other control drug treatments that can be used to compare the efficacy of these peptides which actual approved drug for RCC?
4. Figure 2E-F: Here the difference in IC50 values compared to the cancer cell lines is less than 2-fold difference, how can you count this as being selective only to cancer cells and not epithelia? Clarify on this before concluding.
5. Figure 4: Please quantify the western blots. I do see some decrease in quantities at higher concentration of peptide. Does this show in fold-difference compared to non-peptide treatment?
The overall research is sound, and it needs above additional clarifications or re-doing some of the control experiments in cell viability for it to be accepted.
Reviewer 2 Report
This study detected one peptide derived from HSBP1 as a candidate for targeted therapy against ccRCC through the mass spectrometry-based peptidomic analysis. Moreover, the authors developed Tat-hsbp1 and showed its in vitro efficacy in RCC cell lines.
1) In Introduction (line 42-45), the authors described that "At present, targeted therapy is the main medical management of metastatic ccRCC due to its high resistance to conventional chemo- and radiotherapies, those targeted therapeutic agents mainly targeting the VEGF signaling axis, mTOR pathway and tyrosine kinases.". but the main medical treatment of metastatic ccRCC is immune checkpoint inhibitors. Targeted therapy is used as a combination with immune checkpoint inhibitors or in the favorable group according to the IMDC risk classification.
2) In the Materials and Methods (line 220), the authors described that all data were expressed as mean and SD. But some figures appear to show mean and standard error (SE) because their error bars are relatively short. I think it is necessary to make sure that all data are shown as mean and SD, not mean and SE.
3) It is better to show the baseline characteristics of patients used in peptidomic analysis.
4) In the Results (line 264), was the protein HSPB1 selected based on the result of PPI network analysis? I could not find any comment about the reason why the protein HSPB1 was selected.
5) In Figure 4B, the font of x/y axis is too small and it is difficult to read.
6) In Figure 4C, cleaved-caspase 8 and cleaved-caspase 9 levels after treatment appear to be the same level with the control for me. Please use data showing a more obvious increase. Otherwise, it is necessary to quantitate the results of Westernblot.
7) Do the authors have in vivo therapeutic efficacy of Tat-hsbp1? In vivo mouse data will significantly increase the impact of this study.
Round 2
Reviewer 1 Report
The authors have answered all the questions.
Reviewer 2 Report
The authors answered my questions correctly.